# A General Small-Angle X-ray Scattering-Based Screening Protocol for Studying Physical Stability of Protein Formulations

**DOI:** 10.3390/pharmaceutics14010069

**Published:** 2021-12-28

**Authors:** Fangrong Zhang, Gesa Richter, Benjamin Bourgeois, Emil Spreitzer, Armin Moser, Andreas Keilbach, Petra Kotnik, Tobias Madl

**Affiliations:** 1Key Laboratory of Gastrointestinal Cancer, Ministry of Education, School of Basic Medical Sciences, Fujian Medical University, Fuzhou 350122, China; fangrongzhang@fjmu.edu.cn; 2Gottfried Schatz Research Center for Cell Signaling, Metabolism and Aging, Molecular Biology and Biochemistry, Medical University of Graz, 8010 Graz, Austria; gesa.richter@gmail.com (G.R.); benjamin.bourgeois@medunigraz.at (B.B.); emil.spreitzer@medunigraz.at (E.S.); 3Anton Paar GmbH, 8054 Graz, Austria; armin.moser@anton-paar.com (A.M.); andreas.keilbach@anton-paar.com (A.K.); petra.kotnik@anton-paar.com (P.K.); 4BioTechMed-Graz, 8010 Graz, Austria

**Keywords:** biopharmaceuticals, SAXS, formulation, stability, protein

## Abstract

A fundamental step in developing a protein drug is the selection of a stable storage formulation that ensures efficacy of the drug and inhibits physiochemical degradation or aggregation. Here, we designed and evaluated a general workflow for screening of protein formulations based on small-angle X-ray scattering (SAXS). Our SAXS pipeline combines automated sample handling, temperature control, and fast data analysis and provides protein particle interaction information. SAXS, together with different methods including turbidity analysis, dynamic light scattering (DLS), and SDS-PAGE measurements, were used to obtain different parameters to provide high throughput screenings. Using a set of model proteins and biopharmaceuticals, we show that SAXS is complementary to dynamic light scattering (DLS), which is widely used in biopharmaceutical research and industry. We found that, compared to DLS, SAXS can provide a more sensitive measure for protein particle interactions, such as protein aggregation and repulsion. Moreover, we show that SAXS is compatible with a broader range of buffers, excipients, and protein concentrations and that in situ SAXS provides a sensitive measure for long-term protein stability. This workflow can enable future high-throughput analysis of proteins and biopharmaceuticals and can be integrated with well-established complementary physicochemical analysis pipelines in (biopharmaceutical) research and industry.

## 1. Introduction

Biopharmaceuticals have revolutionized the treatment of a wide range of diseases and are used in almost all branches of medicine [1]. Therapeutic proteins are the fastest growing category of biopharmaceuticals for use in many clinical settings, including cancers, infectious diseases, organ transplantation, chronic inflammatory, and cardiovascular diseases [2]. Biopharmaceutical products represent an increasing percentage in drug development and new drug applications for market approval [3], but their commercial and academic usage is currently limited by their physical stability. In contrast to small-molecule drugs, biopharmaceuticals are potentially immunogenic. Even slight alterations in the structure of the active ingredients in biopharmaceuticals can significantly affect their efficacy and immunogenicity [4,5]. Stabilization of biopharmaceuticals is generally performed during drug development, which involves ensuring not only their proper function but also that their structure is preserved [6]. When the protein is stored in a non-optimal condition, this can lead to degradation or aggregation, which may, in turn, affect the drug’s effectiveness and cause adverse immunological responses [7,8].

Several factors determine physical stability, including concentration, pH, temperature, surfactants, salts, sugars, amino acids, or excipients [9,10]. In addition, the balance between attractive and repulsive interactions between proteins and between proteins and small molecules (additives) can affect protein stability [11]. On the molecular level, driving forces are combinations of hydrogen bonding, hydrophobic and electrostatic interactions [12]. The approaches used most frequently to stabilize proteins include controlling solution pH, surfactants, and co-solvents like amino acids, sugars, and salts in order to suppress protein aggregation, reduce surface adsorption, or simply provide physiological osmolality [13,14]. In these ways, repulsive unspecific protein-protein interactions are considered to be favorable for protein stability [15,16]. Summarizing, identification of suitable conditions for protein purification, storage, and formulation, is a critical step for all protein-based biopharmaceuticals. However, this is often a complex, time-consuming, and cost-intensive effort.

A toolbox of techniques is available for the characterization of protein physicochemical properties, with dynamic light scattering (DLS), size-exclusion HPLC (SE-HPLC), and differential scanning calorimetry (DSC) being the most widely used techniques in biopharmaceutical research and industry. DLS has been widely used for detecting protein aggregation/mean radius. However, DLS is limited by the lower resolution used to distinguish features of polydisperse samples and is susceptible to interference from dispersants [17]. Protein suspensions are often highly heterogeneous and polydisperse and may contain monomers (native, partially unfolded, unfolded), dimers, and oligomers or aggregates [18,19,20,21]. Multiple scattering, a high concentration of small particles whose scattering intensity is interfered with, or a small number of large particles, hamper accurate DLS measurements [22]. It has been shown that SE-HPLC can be used to characterize the composition of therapeutic proteins, mainly for the analysis of purified protein [23,24]. However, SE-HPLC has limited resolution, which can determine the presence of aggregates but not clearly reveal structural changes, and requires a long operating time [25]. DSC provides a thermodynamic profile of the protein, including change in heat capacity (∆Cp), enthalpy (∆H), entropy (∆S), and Gibbs free energy (∆G)) and can be used to assess the structural conformation [26,27]. Its accuracy and sensitivity are limited; for example, only >10% of denatured proteins can be detected [28].

Small-angle X-ray scattering (SAXS) is a robust technique providing insight into the physicochemical properties of biological macromolecules in solution [29]. SAXS is suitable to characterize equilibrium mixtures and dynamic processes, providing structural information through parameters such as the radius of gyration (R_g_) [30,31]. SAXS can be utilized to monitor biomacromolecule conformational changes, protein-protein interactions, assembly states (oligomerization and aggregation), intermolecular attraction and repulsion, and dynamics [32,33,34,35,36]. In situ SAXS studies also allow real-time monitor of the structural changes of proteins [37,38]. Consequently, SAXS could be an extremely beneficial technique for high throughput formulation screening.

In this study, we present a SAXS methodology to investigate proteins in a high-throughput formulation screening setup that can be combined with available information derived from well-established techniques such as DLS and SE-HPLC. We used lysozyme, human serum albumin (HSA), and therapeutic antibody fragments as model proteins to demonstrate and validate our proposed workflow. We sought to create a generalizable analytical workflow that can be applied to a wide range of formulations by SAXS aimed at the identification of optimal formulations and prediction of long-term storage stability. The storage stability progress was followed in situ and in real-time using SAXS. A comprehensive analysis of the physical stability of model protein formulations has been carried out in this study. In situ SAXS-based investigations should be very effective in disclosing biochemical processes accompanied by measurable structural changes.

## 2. Materials and Methods

### 2.1. Materials

Albumin from human serum (10 mg/mL; Sigma, Vienna, Austria) and lysozyme (Applichem, Darmstadt, Germany) were dissolved into ddH_2_O. Antibody fragments were provided by Boehringer Ingelheim RCV GmbH and Co KG (Vienna, Austria). All samples are from the same protein batch, and several methods were applied under the same experimental conditions. The formulation screen was performed at different pH values and buffer stocks (0.5 M concentration) using the JBScreen Buffers (Jena Bioscience, Jena, Germany, Appendix A), where SPG buffer (column 12) is produced by mixing succinic acid:sodium dihydrogen phosphate:glycine in the molar ratio 2:7:7.

### 2.2. Turbidity Assay

Model protein samples (final concentration: 5 mg/mL) were mixed with different buffer/pH from JBScreen Buffers (final concentration: 25 mM from 0.5 M buffer stocks). Turbidity measurements were conducted at 600 nm in 96-well plates with 80 µL samples using a FLUOstar Omega Microplate Reader (BMG Labtech). All experiments were performed in triplicate. When the OD_600 nm_ approached 0.5, the solution showed turbidity by visual inspection. Therefore, we use 0.5 as an OD_600 nm_ threshold for the initial formation of aggregates.

### 2.3. Small-Angle X-ray Scattering

The formulation setup is the same as for the turbidity assay at room temperature (25 °C). SAXS data for model protein formulations were recorded on an in-house SAXS instrument (SAXSpace, Anton Paar, Graz, Austria) equipped with a Kratky camera, a sealed X-ray tube source, and a Mythen2 R 1 K Detector (Dectris). Samples were loaded using the automated sample changer or the sealed sampler loader (in situ analysis). One frame with a 10-min exposure time was measured for each of the different pH/buffers at 5 mg/mL concentrations. A range of momentum transfer of 0.012 < q < 0.63 Å^−1^ was covered (q = 4π sin(θ)/λ, with 2θ the scattering angle and λ = 1.5 Å the X-ray wavelength). All SAXS data were analyzed and processed with the SAXSanalysis package by Anton Paar (version 4.0). We have generated a script (Appendix A) that can be used to combine all SAXS data in one Excel sheet (extract q range, file names, and scattering intensity) and can be used to automatically calculate radius of gyration (R_g_) values for comparison using the following formula (Appendix A):$$R_{g}=\frac{\sqrt{3\times (\ln\left({\;I}_{0}\right)-\ln\left(I\left(q\right)\right)}}{q}$$

The script runs a series of commands and can be executed on Linux/Unix (sub)systems. The SAXS input data must be provided in a format containing three columns (q, intensity, error). The protocol has been deposited at http://smallangle.org/ (accessed on 22 December 2021).

### 2.4. Dynamic Light Scattering

The DLS analyses were carried out using a newly developed instrument, the SpectroLight 610 (XtalConcepts GmbH, Hamburg, Germany). Samples were pipetted onto a 96-well Terasaki plate (Nunclon Delta; catalog No. 1-36528, Nunc GmbH, Wiesbaden, Germany) in volumes of ∼2 µL. Prior to use, the plates were filled with paraffin oil (paraffin oil light; catalog No. A4692, AppliChem, Darmstadt, Germany) to protect the sample solutions from drying out. The laser wavelength used was 660 nm at a power of 100 mW. The scattering angle for the placement of the detector was fixed at 150°. All investigated sample solutions were aqueous; therefore, the refractive index of water (1.33) was used for all calculations. All samples were measured at 293 K.

### 2.5. SDS-PAGE

NuPAGE 4–12% Bis-tris gels, NuPAGE MOPS SDS running buffer, and NuPAGE LDS Sample Buffer (Invitrogen, Milan, Italy) were prepared for SDS PAGE. Prior to electrophoresis, protein samples were denatured by mixing 10 μL of protein solution (8 μg/μL) and 10 μL of NuPAGE sample buffer. The mixture was heated at 80 °C for 5 min to reduce the protein. The reduced protein samples (10 μL) were loaded into the wells of the gels, and electrophoresis was run at a voltage of 200 mV for 50 min. Once the dye front reached the bottom of the gel, the gel was stained with 0.15% Coomassie Brilliant Blue in 50% (*v*/*v*) methanol and 7% acetic acid. After the staining, the gels were destained with a solution composed of 7% acetic acid and 20% methanol. The molecular weights of proteins were determined by comparing the molecular weights of the proteins present in the sample against the protein standards (10–175 kDa, ROTI^®^Mark BI-PINK).

## 3. Results

We developed a generalizable analytical workflow that can be applied to a wide range of formulations by SAXS (Figure 1). A set of 22 different reagents covering a pH range from 5.5 to 8.5 were used for the screening study. These chemicals are frequently used buffer conditions for academic and industrial applications, including five major categories: (i) phosphate; (ii) carboxylic acids (citrate, succinate, malonate, MES, malate, ADA); (iii) amines (Tris and Bis-Tris) and (iv) amino acids (glycylglycine, AMPO, bicine, tricine); (iv) others (imidazole, MOPS, PIPES, DIPSO, TAPS, TAPSO, SPG, HEPES, AMPD). Here, we utilized 84-conditions JBScreen Buffer with a broad range of pH, ionic strength, and additive types. It allows the separation of the influence of the pH and the buffering substance while evaluating the effect of pH. The broad pH ranges and common additives are frequently used in protein purification or storage. Firstly, we carried out a turbidity analysis, where an increase in OD600 indicates an increase in protein size or an aggregation behavior. This was followed by the SAXS experiments, and the R_g_ values were automatically generated by our script. According to the R_g_ value, different colors were used to visualize the R_g_ differences. The color green showed smaller R_g_ values indicating repulsive forces, while larger values indicated aggregation or increasing size (red). We chose water as the reference and selected extreme R_g_ values as the optimal storage condition or the worst storage condition. Prolonged storage in a stable manner of proteins is more challenging for optimal formulations [39]. The accelerated stability studies are typically performed at 40 °C and carried out to predict the aggregation or degradation over prolonged storage periods at standard conditions. Using in situ SAXS, we performed a real-time analysis, recording the changes in R_g_ values over 48 h with 10 min increments. As a complementary assay, we also used SDS-PAGE to monitor protein degradation before and after storage at 40 °C for 48 h. In addition, DLS has been used to evaluate aggregation behavior in the same conditions. Here, a comprehensive study of formulation screening has been thoroughly studied, integrating the different techniques and at the same time allowing us to compare the differences among techniques.

### 3.1. Lysozyme as a Model Protein

This study utilized lysozyme as a model protein, a 14.3 kDa basic protein, which has provided detailed properties and reasonable insights into its biological activity [40,41,42,43]. It is still unclear whether intrinsic properties of proteins are associated with solubility and stability, so systematic screening is necessary to identify optimal conditions for samples. As shown in Figure 2A, the rise in OD600 of lysozyme has been observed in AMPD buffer with pH 8 (OD600: 0.456) and 8.5 (OD600:0.436), indicating an increase in the size and/or formation of aggregates. Compared to the OD600 turbidity study, SAXS seemed to show multi-layers and more sensitive results (Figure 2B). The higher R_g_ values are consistent with higher OD600 values from the turbidity analysis. Notably, the turbidity analysis did not monitor the changes among some conditions that could be observed in SAXS. Lysozyme shows extremely basic pI (around 11), and an increased R_g_ value can be investigated at pH > 8, which may be due to the solubility of proteins being minimal at pH solution conditions close to their pI [44]. In addition, lysozyme disfavors the storage condition as an SPG buffer. Here, H_2_O (as reference); bis-tris propane buffer, pH 7.0; phosphate buffer, pH 8.0; TAPS buffer, pH 8.5 were selected as extreme buffer conditions for the next step in the accelerated stability study (Figure 2C). Lysozyme with H_2_O and bis-tris propane buffer underwent structural changes at 40 °C over time, as indicated by increasing R_g_. The sample in the most destabilizing buffer conditions (phosphate buffer, pH 8.0; TAPS buffer, pH 8.5) exhibited a higher invariable R_g_ value, suggesting that aggregation-like behavior occurs from the beginning in these buffers. The optimal buffer condition (bis-tris propane buffer, pH 7.0) was observed to yield the smallest R_g_ up to the maximal duration of 48 h compared to the other buffers tested. Mean radii were measured by DLS with these buffer conditions before and after 48 h storage at 40 °C to understand protein conformational stability of lysozyme. All buffer conditions showed a minor impact at the first time point since the mean radii of lysozyme were similar. After incubation at 40 °C for 48 h, bis-tris propane buffer seemed more effective as an additive to prevent aggregation, whereas the mean radius of lysozyme significantly increased with other additives (Figure 2D). Data in Figure 2E suggests that degradation did not occur in all buffers evenly after 48 h of heating (40 °C).

### 3.2. HSA as a Model Protein

HSA is a monomeric 66.5 kDa protein synthesized by the liver. It represents the most abundant protein in the blood serum and associates with many substances consisting of hormones or drug processes [45]. HSA can form well-defined aggregates: dimers, oligomers, and even larger structures [46,47,48,49]. HSA is a well-studied and highly-available protein and was therefore selected as a model protein. Turbidity analysis did not observe a very significant aggregation behavior (Figure 3A), but the results in SAXS showed more pronounced variations depending on the buffer conditions (Figure 3B). Citrate buffer (pH < 7), glycylglycine buffer, and imidazole buffer (pH < 7) can be satisfactory protein aggregation suppressors. The three extreme conditions (phosphate, pH 8.0; citrate, pH 7.0; TAPS, pH 7.7) and water have been chosen for 48 h storage at 40 °C, and SAXS data were recorded every 10 min. The time-dependent increase in R_g_ detected by SAXS for HSA may reflect a shift in population from monomer to dimer or aggregate formation in H_2_O, phosphate pH 8.0, and citrate pH 7.0. Slight changes of R_g_ of HSA were detected in TAPS, pH 7.7 buffer, indicating that HSA is stable in this buffer (Figure 3C). The mean radius of HSA with different buffer conditions was determined by DLS measurements (Figure 3D). At the initial time point, HSA showed the same mean radius in all buffer conditions tested. The resulting DLS distribution histogram indicated an increase in both the amount and mean radius of aggregated particles for HSA with different buffer conditions after 48 h storage at 40 °C. The effects of temperature on the aggregation of HSA have been studied, and the dependence of structural alterations is correlated with free —SH groups at thermal denaturation [50]. Heat treatment raises the proportion of β structures, which is relevant to the aggregation of HSA [51]. Here, the increased mean radius of HSA at high temperatures further illustrates its temperature sensitivity. The control SDS-PAGE profiles are shown in Figure 3E. Proteolytic degradation has not been observed in all buffer conditions before and after 48 h storage at 40 °C.

### 3.3. Therapeutic Antibody Fragment as a Model Protein

The therapeutic antibody fragment was provided by Boehringer Ingelheim RCV GmbH and Co KG (Vienna, Austria). The antibody fragment is a highly water-soluble trimeric protein with a molecular weight of 44.4 kDa, consisting of 439 amino acid residues. Here, the buffer screen enables an extensive systematic comparative analysis of different additives, pH, and temperature for this biopharmaceutical model protein. The maximum OD600 for the antibody fragment was 0.60 in DPSD buffer, pH 8.0, and elevated OD600 of 0.34 and 0.36 were detected for samples in DPSD buffer, pH 7.5 and bicine, pH 7.5 compared to other buffer conditions, respectively (Figure 4A). SAXS data resolved more differences among buffers compared to turbidity analysis. Overall, the antibody fragment seems to generally disfavor extreme acidic or basic buffer conditions. The highest and second highest R_g_s values were observed in PIPES pH 6.1 and imidazole pH 7.5, and therefore we selected these two as the worst storage conditions. AMPD pH 8.5 was selected as an optimal buffer condition for further stability studies. In accordance with the observed R_g_ screen results of additives for the antibody fragment, increasing aggregation (R_g_) was observed in PIPES pH 6.1 and imidazole pH 7.5 buffer conditions. In contrast, only slight changes were observed after 48 h 40 °C storage in AMPD pH8.5. R_g_s values were also increased in H_2_O; hence, aggregation was most pronounced in the absence of any additives during 48 h 40 °C storage compared to the worse storage conditions. DLS data showed the most obvious resistance of the antibody fragment in AMPD pH 8.5 to aggregation at the initial time point. All selected buffer conditions show aggregation after 48 h at 40 °C storage. Using the same experimental conditions as for the DLS analysis, degradation has not been detected by SDS-PAGE (Figure 4E).

## 4. Discussion

According to a report by Mordor Intelligence, the biopharmaceuticals market was USD 325.17 billion in 2020. It has been estimated that the revenue will grow up to USD 496.71 billion in 2026. In contrast to small molecules, biopharmaceuticals are notoriously sensitive to manufacturing processes, starting materials, and storage conditions [52]. The development of biopharmaceuticals involves extensive physical stability characterizations which require intensive labor and costs. The selection of a suitable storage environment is crucial for the biopharmaceuticals’ physical stability and efficacy [53]. Currently, the most common and largest class of biopharmaceuticals is therapeutic proteins [54]. Buffers or additives are selected to minimize the self-association of proteins and thus prevent aggregation while ensuring that the protein structure is not altered [55]. An optimized formulation condition (pH, buffer, ionic strength) can be used to suppress the formation of protein aggregates and preserve therapeutic function [56,57]. Providing a robust formulation screening strategy would help reduce costs and make biologic therapies affordable. Consequently, it is important to establish a fast, robust, and highly automated characterization strategy for physical stability.

In this study, we have setup a pipeline for the analysis of protein aggregation by using laboratory SAXS as a key technique. The physical stability of therapeutic protein, i.e., lysozyme, HSA, and antibody fragments, were analyzed using standard techniques like light spectroscopy, DLS, and SDS-PAGE. DLS can qualitatively detect aggregates and offer long-term measurements by comparing the mean radii in different formulations/time points and has been applied widely [58,59,60,61]. Based on the observations here, turbidity analysis can be used as a preliminary assessment of stability. After checking the formation of visible aggregation by measuring absorption at 600 nm, we propose to use laboratory SAXS to detect particle interactions, such as particle repulsion and formation of soluble aggregates. Comparing OD600 measurements, SAXS, and DLS, we found that SAXS is more sensitive in detecting aggregation than light spectroscopy and DLS.

In situ SAXS helps to improve real-time monitoring of protein conformational changes or turnover time points complementing the DLS analysis [43]. We found that SAXS measurements can provide valuable hints for the prediction of long-term storage stability, as higher radii of gyration correlated with poor stability in buffers in accelerated aging experiments at 40 °C. Moreover, different proteins showed different sensitivities to temperature. For example, lysozyme has higher conformational stability at 40 °C in optimal buffers, which is consistent with DLS data and DLS-Raman [62]. In a suitable buffer, lysozyme activity seems to be largely preserved below 60 °C [63,64]. HSA is temperature sensitive and tends to form aggregates at elevated temperatures [65]. Correspondingly, proteins exhibited different sensitivity to additives and pH. Low concentrations of TAPS (<0.5 M) have been reported to preserve the secondary structure of HSA, which is in line with our result [66]. pH is thought to be used as a chemical stressor, with extreme acidic or basic environments triggering the formation of aggregates [67]. pH 7 presented the strongest lysozyme activity in bis-tris propane buffer, which is consistent with our optimal storage conditions found for lysozyme [68]. To check for proteolytic degradation, we used SDS-PAGE, a method commonly used in the biopharmaceutical industry [69,70]. The different buffers, as well as the storage at 40 °C, did not induce protein degradation as observed by SDS-PAGE for all three model proteins.

Despite its rapidly growing use in biomedicine, SAXS is not yet the standard method for formulation screening in the biopharmaceutical industry [35,71]. This is primarily due to the fact that most SAXS studies are carried out using synchrotron SAXS, which limits its availability in the biopharmaceutical industry. The recent improvements in affordable laboratory SAXS instrumentation have made it possible to investigate biomolecular structure and dynamics in-house [72,73,74,75,76,77,78] and therefore enabled a plethora of additional possibilities for SAXS, including SAXS analysis of biopharmaceuticals. With the use of autosamplers, laboratory SAXS can easily deal with hundreds of samples in a short period, which makes it highly suitable for high-throughput screening [29,79]. In the current study, we used laboratory SAXS to screen for optimal protein buffer conditions using a simple parameter, R_g_, as a read-out. In line with results from turbidity experiments, DLS, and SDS-PAGE, SAXS measurements indicate similar aggregation behaviors with protein in specific buffers.

## 5. Limitation

Fast aggregation processes might not be picked up by SAXS due to the required measurement times. The development of sample cells with rapid mixing combined with in situ SAXS might help to overcome these limitations in the future. Alternatively, FPLC-SAXS might be used to reduce the time difference for immediate analysis [80]. Here, we do not provide further structural information for more advanced data analyses. For example, conformational differences in IgG in different solutions have been revealed [81,82]. Here, we have performed turbidity and SAXS analyses on all buffer conditions to compare various methods. The prepared 96-well plates from the same sample can first be subjected to turbidity analysis to exclude some buffers that already show aggregation behavior to decrease the analyses time of the SAXS screen. In its current implementation, the analysis script can be executed on Unix/Linus subsystems. In the future, implementation of the pipeline in SAXS analysis software would be desired.

## 6. Conclusions

SAXS can provide an effective tool for formulation screening, strongly supporting the selection and development of formulations for biopharmaceuticals. By using an automated setup, high throughput analysis of up to 192 samples can be achieved. For the model proteins tested in this study, SAXS was more sensitive for protein-protein interactions or conformational changes under different formulation conditions, and these differences correlated with protein stability in accelerated stability studies. Together with a straightforward analysis, this will facilitate the development of SAXS as a rapid screening method for formulation development. In addition, ongoing developments in SAXS instrumentations, such as high-flux MetalJet X-ray sources and low-volume autosamplers, may further facilitate [83,84,85] the establishment of SAXS as a key technique in biopharmaceutical research and industry in the near future. It can be envisioned that SAXS big datasets obtained from screenings described here, together with sequence and structural information, could be a useful database for training AI algorithms in the future. First studies have demonstrated the synergy of SAXS and machine learning to predict the physical properties of biomolecules based on SAXS data [86,87,88,89]. Our pipeline can provide high-throughput SAXS datasets as a function of the plethora of variables such as pH, concentration, temperature, surfactants, salts, sugars, amino acids, or excipients and may be used to predict physical stability via machine learning in the future.

## Figures and Tables

**Figure 1 pharmaceutics-14-00069-f001:**
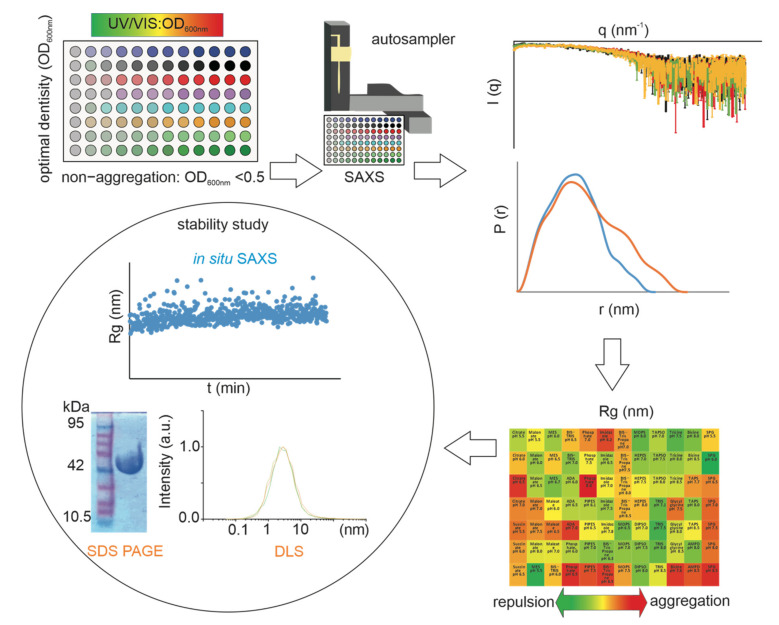
Illustration of the SAXS Screen workflow on a single screening for studying physical stability of protein formulations.

**Figure 2 pharmaceutics-14-00069-f002:**
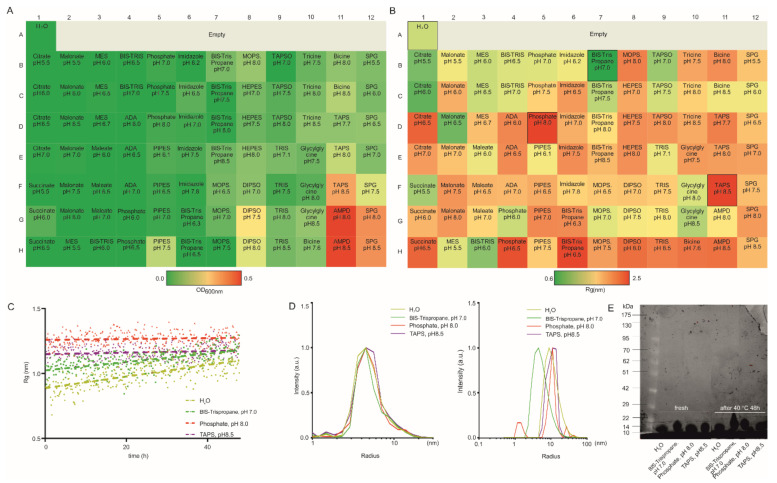
(**A**) A microplate assay for measuring OD 600 nm of lysozyme (5 mg/mL) colored by value (>0.5 corresponds to aggregation). (**B**) SAXS-based screening of lysozyme (5 mg/mL) colored by the radius of gyration (R_g_). (**C**) Changes in R_g_ were obtained for lysozyme at extreme formulations with 40 °C. (**D**) Mean radii distribution of lysozyme at extreme formulations with 40 °C before and after 48 h measured by DLS. (**E**) SDS-polyacrylamide gel electrophoresis (SDS-PAGE) of lysozyme incubated at extreme formulations with 40 °C before and after 48 h.

**Figure 3 pharmaceutics-14-00069-f003:**
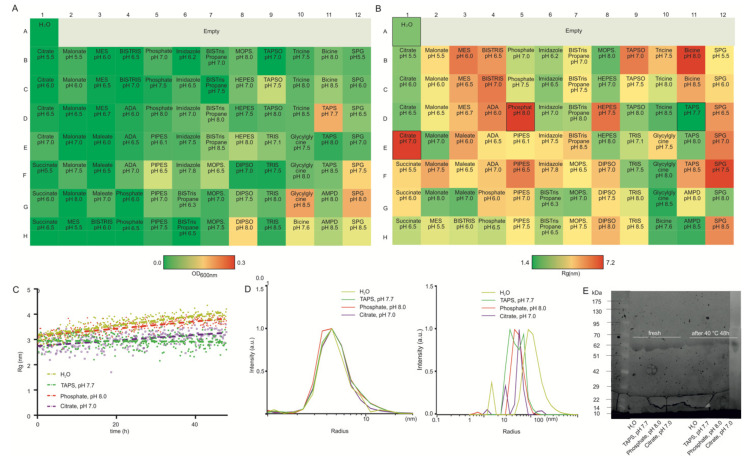
(**A**) A microplate assay for measuring OD 600 nm of HSA (5 mg/mL) colored by value (>0.5 means aggregation). (**B**) SAXS-based screening of HSA (5 mg/mL) colored by the value of the radius of gyration R_g_. (**C**) Changes in R_g_ value were obtained for HSA at extreme formulations with 40 °C. (**D**) Mean radii distribution of HSA at extreme formulations with 40 °C before and after 48 h measured by DLS. (**E**) SDS-polyacrylamide gel electrophoresis (SDS-PAGE) of HSA incubated at extreme formulations with 40 °C before and after 48 h.

**Figure 4 pharmaceutics-14-00069-f004:**
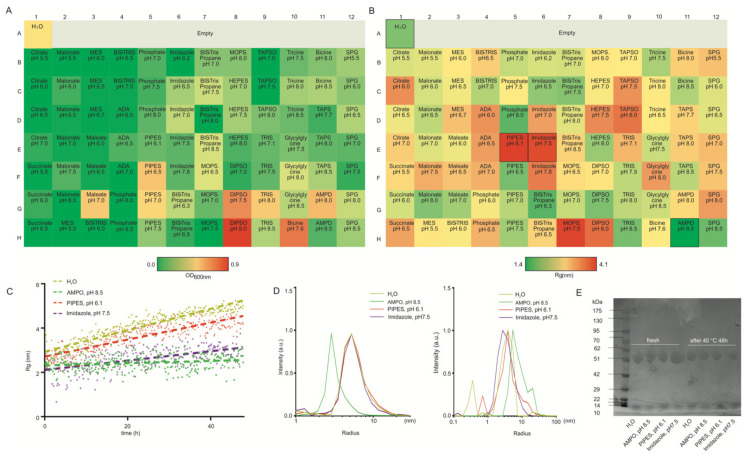
(**A**) A microplate assay for measuring OD 600 screening of the antibody fragment (5 mg/mL) colored by value (>0.5 means aggregation). (**B**) SAXS-based screening of the antibody fragment (5 mg/mL) colored by the value of the radius of gyration R_g_. (**C**) Changes in Rg value were obtained for the antibody fragment at extreme formulations with 40 °C. (**D**) Mean radii distribution of the antibody fragment at extreme formulations with 40 °C before and after 48 h measured by DLS. (**E**) SDS-polyacrylamide gel electrophoresis (SDS-PAGE) of the antibody fragment was incubated at extreme formulations at 40 °C before and after 48 h.

## Data Availability

The data presented in this study are available on reasonable request from the corresponding author.

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
