# Peer review of "A General Small-Angle X-ray Scattering-Based Screening Protocol for Studying Physical Stability of Protein Formulations"

_pharmaceutics, 2021, doi:10.3390/pharmaceutics14010069_

Round 1

Reviewer 1 Report

The review of a manuscript entitled "A general small-angle X-ray scattering-based screening protocol for studying physical stability of protein formulations" by Fangrong Zhang.

The aim of the presented study was to design and evaluate a general workflow for screening of protein formulations based on small-angle X-ray scattering (SAXS) technique. The authors compared the results from SAXS with other well established methods such as turbidity analysis, dynamic light scattering and SDS-PAGE measurements.

in the introduction, the authors clearly defined the benefits of developing a new rapid screening for the physical stability of protein systems. The introduction does not overwhelm with information and is rather an invitation to read the article further. The methodology is clearly described and sufficiently to replicate the experiment in your own laboratory. Nevertheless, the question arises regarding data processing, will the script presented by the authors be applicable to data received from SAXS from another vendor? It should also be mentioned that the written script, while not very complicated, can only be run under the csh shell on Unix systems. Except the script part, which need to be somehow extended, the results part is clearly described and the discussion section is pin-pointing strong sides of the research. Nevertheless, the authors see also limitations of this technique, which were described in the section "Limitations".

Overall, I highly rate the work and recommend it for publication in Pharmaceutics MDPI. I would like to see more results based on the pipeline (a validation, maybe?).

Author Response

The review of a manuscript entitled "A general small-angle X-ray scattering-based screening protocol for studying physical stability of protein formulations" by Fangrong Zhang.

The aim of the presented study was to design and evaluate a general workflow for screening of protein formulations based on small-angle X-ray scattering (SAXS) technique. The authors compared the results from SAXS with other well established methods such as turbidity analysis, dynamic light scattering and SDS-PAGE measurements.

in the introduction, the authors clearly defined the benefits of developing a new rapid screening for the physical stability of protein systems. The introduction does not overwhelm with information and is rather an invitation to read the article further. The methodology is clearly described and sufficiently to replicate the experiment in your own laboratory.

We thank Reviewer 1 for her/his positive evaluation of our manuscript.

1) Nevertheless, the question arises regarding data processing, will the script presented by the authors be applicable to data received from SAXS from another vendor?

The script can be applied to SAXS data received from any vendor as it requires only “raw” SAXS profiles containing a 3 column format (q, Intensity, Error). We have added a corresponding statement to the revised version of the manuscript.

2) It should also be mentioned that the written script, while not very complicated, can only be run under the csh shell on Unix systems. Except the script part, which need to be somehow extended, the results part is clearly described and the discussion section is pin-pointing strong sides of the research. Nevertheless, the authors see also limitations of this technique, which were described in the section "Limitations".

Overall, I highly rate the work and recommend it for publication in Pharmaceutics MDPI. I would like to see more results based on the pipeline (a validation, maybe?).

We thank Reviewer 1 for her/his suggestion. As suggested, we updated the text accordingly.

The Script is performed by issuing a series of commands to an interpreter, and can only be run on Linux/Unix (sub)systems. Indeed, we plan to submit a follow-up study.

Reviewer 2 Report

The manuscript "A general small-angle X-ray scattering-based screening protocol for studying physical stability of protein formulations" presents a screening method performed by SAXS and DLS in order to characterize the stability of the proteins in terms of formulation (such as pH, concentration, temperature, surfactants, salts, sugars, amino acids, excipients etc.) and in the course of time. The authors selected 3 examples of standard proteins to demonstrate the feasibility, namely lysozyme, HSA, and an antibody fragment. The authors claim that DLS is not sensitive enough to fully support such a study, while SAXS is much more suitable for that purpose. The author s focus on the radius of gyration, Rg, as the key magnitude in focus of the SAXS characterization. In this sense a classical Kratky camera is fully sufficient. So the authors define a protocol for stability measurements that is highly desired for pharmaceutical research, and do not aim at top-end protein characterization where the full SAXS curve is analyzed in order to get many more details about the structure of proteins, although the PDF p(r) is calculated but seems to be omitted. I rate this manuscript highly valuable for industrial research. So I would agree with the publication of the manuscript - maybe with some minor corrections.

Minor points:

The radius of gyration has some experimental error. Does this error enter the screening protocol in a reasonable way?

The canSAS working group (https://www.cansas.net) would be highly interested in such protocols. Maybe a link can be made...!?

Maybe a comment about big data can be made: Which data and in which way is stored? Would the data be made accessible to AI at some point? What is the current status of automatization?

Author Response

The manuscript "A general small-angle X-ray scattering-based screening protocol for studying physical stability of protein formulations" presents a screening method performed by SAXS and DLS in order to characterize the stability of the proteins in terms of formulation (such as etc.) and in the course of time. The authors selected 3 examples of standard proteins to demonstrate the feasibility, namely lysozyme, HSA, and an antibody fragment. The authors claim that DLS is not sensitive enough to fully supporsuch a study, while SAXS is much more suitable for that purpose. The author s focus on the radius of gyration, Rg, as the key magnitude in focus of the SAXS characterization. In this sense a classical Kratky camera is fully sufficient. So the authors define a protocol for stability measurements that is highly desired for pharmaceutical research, and do not aim at top-end protein characterization where the full SAXS curve is analyzed in order to get many more details about the structure of proteins, although the PDF p(r) is calculated but seems to be omitted. I rate this manuscript highly valuable for industrial research. So I would agree with the publication of the manuscript - maybe with some minor corrections.

We thank Reviewer 2 for her/his positive evaluation of our manuscript.

1) The radius of gyration has some experimental error. Does this error enter the screening protocol in a reasonable way?

In its current version, the screening protocol does not include an experimental error of the radius of gyration. Thank you for your suggestion, we will include this in a follow-up version. We checked one screen for typical errors manually using ATSAS and found that the Rg determination is below a 5% error range.

2) The canSAS working group (https://www.cansas.net) would be highly interested in such protocols. Maybe a link can be made...!?

We thank the Reviewer 2 for her/his suggestions. We will contact canSAS working group to deposit our protocol. In the manuscript we now refer to the new website of the canSAS working group http://smallangle.org/.

3) Maybe a comment about big data can be made: Which data and in which way is stored? Would the data be made accessible to AI at some point? What is the current status of automatization?

We have extended the conclusion accordingly and added the following sentences:

“It can be envisioned that SAXS big data sets obtained from screenings described here together with sequence and structural information could be a useful data base for training AI algorithms in the future. First studies have demonstrated synergy of SAXS and machine learning to predict physical properties of biomolecules based on SAXS data (1-4). Our pipeline can provide high-throughput SAXS datasets as function on a plethora of variables such as pH, concentration, temperature, surfactants, salts, sugars, amino acids, or excipients and may be used to predict physical stability via machine learning in the future.”

With the implementation of the analysis pipeline in SAXS software, and using pipetting robots, the entire pipeline can be run fully automatically.

  1. Scherdel C, Miller E, Reichenauer G, Schmitt J. Advances in the Development of Sol-Gel Materials Combining Small-Angle X-ray Scattering (SAXS) and Machine Learning (ML). Processes. 2021;9(4):672.
  2. Franke D, Jeffries CM, Svergun DI. Machine learning methods for X-ray scattering data analysis from biomacromolecular solutions. Biophysical journal. 2018;114(11):2485-92.
  3. Do C, Chen W-R, Lee S. Small angle scattering data analysis assisted by machine learning methods. MRS Advances. 2020;5(29):1577-84.
  4. Demerdash O, Shrestha UR, Petridis L, Smith JC, Mitchell JC, Ramanathan A. Using small-angle scattering data and parametric machine learning to optimize force field parameters for intrinsically disordered proteins. Frontiers in molecular biosciences. 2019;6:64.